# Neonatal health care costs of very preterm babies in England: a retrospective analysis of a national birth cohort

Miaoqing Yang,[1,2] Helen Campbell  ,[1] Thillagavathie Pillay  ,[3,4] Elaine M Boyle,[5] Neena Modi,[6] Oliver Rivero-Arias  [1]

## ABSTRACT

**Objectives** Babies born between $27^{+0}$ and $31^{+6}$ weeks of gestation represent the largest group of very preterm babies requiring National Health Service (NHS) care; however, up-to-date, cost figures for the UK are not currently available. This study estimates neonatal costs to hospital discharge for this group of very preterm babies in England.

**Design** Retrospective analysis of resource use data recorded within the National Neonatal Research Database.

**Setting** Neonatal units in England.

**Patients** Babies born between $27^{+0}$ and $31^{+6}$ weeks of gestation in England and discharged from a neonatal unit between 2014 and 2018.

**Main outcome measures** Days receiving different levels of neonatal care were costed, along with other specialised clinical activities. Mean resource use and costs per baby are presented by gestational age at birth, along with total costs for the cohort.

**Results** Based on data for 28 154 very preterm babies, the annual total costs of neonatal care were estimated to be £262 million, with 96% of costs attributable to routine daily care provided by units. The mean (SD) total cost per baby of this routine care varied by gestational age at birth; £75 594 (£34 874) at 27 weeks as compared with £27 401 (£14 947) at 31 weeks.

**Conclusions** Neonatal healthcare costs for very preterm babies vary substantially by gestational age at birth. The findings presented here are a useful resource to stakeholders including NHS managers, clinicians, researchers and policymakers.

## INTRODUCTION

Babies born between $27^{+0}$ and $31^{+6}$ weeks of gestation (hereafter called 'born at 27–31 weeks') represent the largest group of very preterm babies requiring National Health Service (NHS) care.[1] These babies also account for about 12% of all viable preterm babies born in England and usually require admission to a neonatal unit.[1]

Previous work in the UK and elsewhere has attempted to estimate the healthcare and societal cost of preterm birth.[2–7] However, there is marked variability in

### WHAT IS ALREADY KNOWN ON THIS TOPIC
⇒ Existing cost estimates for very preterm care in England are now over a decade old and may not reflect modern care practices.

### WHAT THIS STUDY ADDS
⇒ This study has generated current cost estimates at the level of the individual baby and for the cohort as a whole in England and confirmed the previously reported inverse relationship between healthcare costs and gestational age at birth.

### HOW THIS STUDY MIGHT AFFECT RESEARCH, PRACTICE OR POLICY
⇒ The outputs from this work provide a valuable resource for research assessing the economic implications of interventions to prevent preterm birth, the provision of care for preterm babies, as well as helping inform National Health Service resource allocation decisions.

reported cost estimates, due to differences in study perspectives, included babies, data sources and methods used to assign costs. A number of studies have reported cost estimates associated with the initial period of hospitalisation for babies born at 27–31 weeks.[2 3 6] While these studies have shown the costs of neonatal care for very preterm babies to be inversely related to gestational age at birth, estimates for the UK are now over a decade old and there is a need for new analyses. As part of the OPTImising neonatal service provision for PREterM babies born between 27 and 31 weeks gestation in England (OPTI-PREM) suite of studies aimed at optimising neonatal service provision for very preterm babies in England, we conducted a retrospective cohort study to describe the levels of neonatal care, key specialist procedures and healthcare costs attributable to the management of these babies.[1] The analysis makes use of healthcare resource use data routinely collated within the National

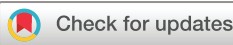

For numbered affiliations see end of article.

**Correspondence to**
Dr Oliver Rivero-Arias; oliver.rivero@npeu.ox.ac.uk

Neonatal Research Database (NNRD) on babies admitted to all neonatal units in England.

## METHODS

### Study design and population

For this retrospective analysis, we included all admissions to neonatal units in England for babies born at 27–31 weeks who were discharged or died between January 2014 and December 2018. Admissions were identifiable through the NNRD, which was created in 2007 to support activities including audit, evaluations and clinical, health services and policy research and is maintained at the Neonatal Data Analysis Unit at Imperial College London (https://www.imperial.ac.uk/neonatal-data-analysis-unit/). All NHS neonatal units in England, Scotland, Wales and the Isle of Man use their Electronic Patient Record (EPR) supplier systems to routinely submit detailed information to the NNRD on the clinical care they provide to babies. Submitted data are quality assured and curated to a research standard. Data comprise demographics, diagnoses, health outcomes and daily interventions (including the level of care, namely intensive, high dependency, special or normal, provided each day) and treatments administered during the inpatient episode.

Data were extracted from the NNRD on 5 October 2020. Daily intervention data and treatment episode data were received in two separate data sets and were linked by means of participant identification number. Data were imported into Stata software and manipulated such that for each baby, a daily intervention record (which included the level of care provided) was available for each day of the admission episode. Babies missing daily intervention records, or for whom daily records were available, but level of care data had not been recorded, were excluded from the primary analysis.

### Resource use and costs

The cost analysis was conducted from the perspective of the NHS in England. To calculate the costs associated with the routine daily neonatal care received by each baby, we multiplied the number of days spent receiving each level of care, by level-specific national average bed day costs sourced from the 2018/2019 National Schedule of NHS Costs.[8] Such costs are estimated by assigning each level of care a healthcare resource group (HRG) code. HRGs are groupings of clinically meaningful activities made primarily on the basis of diagnosis and procedure codes, and within the NHS, are the 'units' of healthcare for which providers receive payment. Costs assigned to each HRG are based on nationally estimated tariffs developed to adequately cover the cost of providing high-quality and cost-effective care. For this analysis, the neonatal bed day costs used were: intensive care (HRG XA01Z), high dependency care (HRG XA02Z), special care without carer resident alongside baby (HRG XA03Z), special care with carer resident alongside baby (HRG XA04Z) and normal care (HRG XA05Z). Online supplemental

table A1 shows these unit costs. Detailed information on the items included within each daily level of care cost is available from NHS England.[9]

Not all major clinical activities are included in these neonatal critical care HRGs. After in-depth discussions with clinical experts, we identified five high-cost non-routine procedures captured within the NNRD, that were considered to be important in this particular population. These were nitric oxide, surfactant replacement, total parental nutrition (TPN) (over 14 days of use), palivizumab use and surgical care. Days receiving nitric oxide and TPN (beyond 14 days) were costed using *per diem* unit costs. Surfactant replacement and palivizumab were costed when given using unit costs specific to a baby's weight. Types of neonatal surgery were identified from their corresponding Operating Procedure Code Standard (OPCS) and the International Statistical Classification of Diseases and Health Related Problems (ICD-10) codes for costing purposes and unit costs obtained from the 2018/2019 National Schedule of NHS Costs. Unit costs for these non-routine procedures are detailed in online supplemental table A1. NNRD data for these five non-routine procedures were assumed to be complete, with no entry in the corresponding data set fields taken as an indication that the procedure did not take place.

### Statistical analysis

When describing neonatal and maternal characteristics, counts and proportions and means and SD were used for categorical and continuous variables, respectively. Analyses of resource use data and costs were performed by gestational age at birth. Means (SD) were used to summarise days provided at each level of care with associated 95% CI following a Poisson distribution. Counts were made of the numbers of babies for whom non-routine procedures were recorded, along with the number of cases (eg, surgeries) or days (eg, for nitric oxide) for which such treatment was given. Total costs were reported for the cohort as a whole and again by gestational age at birth. A secondary analysis used multiple imputation to impute routine care costs at each level (intensive care, high dependency care, etc) for babies with missing daily record and level of care data (see online supplemental file, supplementary Multiple Imputation Methods). Analyses were conducted using Stata MP.[10]

### Patient and public involvement

An OPTI-PREM parent panel of mothers and fathers of babies born at 27–31 weeks in England was established with support from the national charity Bliss. The parent panel engaged in the study design and review of the funding protocol, development of parent information leaflets and neonatal unit posters. They attended team and study steering committee meetings, provided input at national stakeholder discussions and contributed to the interpretation of final results and our dissemination strategy.

**Table 1** Baby and maternal characteristics of study cohort—data are frequencies (percentages) unless otherwise stated

| | Cohort (n=28 173) |
| --- | --- |
| | n (%) |
| Baby characteristics | |
| Gestational age at birth | |
| 27 weeks | 3296 (11.7%) |
| 28 weeks | 4370 (15.5%) |
| 29 weeks | 5036 (17.9%) |
| 30 weeks | 6625 (23.5%) |
| 31 weeks | 8827 (31.3%) |
| Missing | 19 (0.1%) |
| Gender of baby | |
| Male | 15 363 (54.5%) |
| Female | 12 755 (45.3%) |
| Missing | 55 (0.2%) |
| Number of fetuses | |
| Singleton birth | 20 555 (73.0%) |
| Multiple birth | 7598 (27.0%) |
| Missing | 20 (0.1%) |
| Birth weight (g)—mean (SD) | 1330.2 (332.2) |
| Missing | 82 (0.3%) |
| Apgar score at 5 min—mean (SD) | 8.1 (1.8) |
| Missing | 2877 (10.2%) |
| Died in neonatal care | 985 (3.5%) |
| Missing | 0 (0.0%) |
| Maternal characteristics | |
| Age (years)—mean (SD) | 30.7 (6.3) |
| Missing | 279 (1.0%) |
| Ethnicity | |
| White | 17 647 (62.6%) |
| Black | 2011 (7.1%) |
| Asian | 3144 (11.2%) |
| Mixed | 433 (1.5%) |
| Other | 483 (1.7%) |
| Missing | 4455 (15.8%) |
| Diabetes | 2378 (8.4%) |
| Missing | 10 736 (38.1%) |
| Hypertension | 3506 (12.4%) |
| Missing | 10 460 (37.1%) |
| Infection | 3029 (10.6%) |
| Missing | 10 595 (37.6%) |
| Mode of delivery | |
| Vaginal spontaneous | 8200 (29.1%) |
| Vaginal instrumental | 765 (2.7%) |
| Caesarean section | 17 691 (62.3%) |
| Missing | 1517 (5.4%) |

Continued

**Table 1** Continued

| | Cohort (n=28 173) |
| --- | --- |
| | n (%) |
| Quintiles of IMD | |
| First Q, least deprived | 3355 (11.9%) |
| Second Q | 3761 (13.4%) |
| Third Q | 4516 (16.0%) |
| Fourth Q | 5930 (21.1%) |
| Fifth Q, most deprived | 8057 (28.6%) |
| Missing | 2554 (9.1%) |

IMD, Index of Multiple Deprivation; Q, quintile.

## RESULTS

### Study population

Within the NNRD data extraction were 29 842 infants, with a total of 1 512 446 daily records and 46 746 episode records. Data were available from all but one neonatal unit in England. Following the removal of infants with missing daily record information (n=1292) and those with missing data on the level of daily care provided (n=377), a total of 28 173 babies (94% of the starting cohort of 29,842) remained and were included in the cost analysis (full details of the record matching process and data cleaning is found in online supplemental figure A1). Table 1 summarises the characteristics of these babies and their mothers. A comparison between babies included in the analysis and those excluded due to missing data (n=1669) was performed and showed the latter were more likely to have been born at earlier gestations (eg, 17% vs 12% were born at 27 weeks; see online supplemental table A2). Despite this, birth statistics for the study cohort were still comparable to those of all very preterm babies born in England between 2016 and 2018 (data compiled by the Office for National Statistics; see online supplemental table A3).

### Resource use

Table 2 shows the mean (SD) per baby duration (days) spent receiving each level of care according to gestational age at birth. Data show a consistent inverse relationship between gestational age at birth and the intensity of daily care provided. Figure 1 plots the mean durations along with mean overall length of stay on the neonatal unit and illustrates the longer durations of higher intensity care (and indeed overall neonatal care) provided to babies born at earlier gestations. For example, babies born at 27 weeks spent on average, 18.1 days (SD=15.7 days) receiving intensive care and 26.8 days (SD=22 days) receiving high-dependency care, while babies born at 31 weeks received an average of 3.33 days (SD=6.7 days) of intensive care and 5.1 days (SD=7.5 days) of high-dependency care. Also of note is that across all gestational age at birth groups, days spent receiving special care without a carer present

**Table 2** Summary of type and duration of daily care received by very preterm babies during neonatal unit admissions in England for the period 2014–2018 (n=28 154)*

| Gestational age at birth | Level of care provided (HRG Code) | | | | |
| --- | --- | --- | --- | --- | --- |
| | Intensive care (XA01Z) | High-dependency care (XA02Z) | Special care without carer (XA03Z) | Special care with carer (XA04Z) | Normal care (XA05Z) |
| **27 weeks gestation (n=3296)** | | | | | |
| Number of days of care | 59 494 | 88 212 | 100 770 | 5113 | 0 |
| Number of babies receiving care | 3275 | 3065 | 3021 | 1681 | 0 |
| Mean (SD) duration of care (days)† | 18.1 (15.7) | 26.8 (22.0) | 30.5 (18.0) | 1.5 (2.5) | 0.0 (0.0) |
| 95% CI | 17.9 to 18.2 | 26.6 to 27.0 | 30.4 to 30.7 | 1.5 to 1.6 | 0.0 to 0.0 |
| Median (range) duration of care (days)† | 14.0 (0.0 to 174.0) | 23.0 (0.0 to 243.0) | 30.5 (0.0 to 161.0) | 1.0 (0.0 to 31.0) | 0.0 (0.0 to 0.0) |
| **28 weeks gestation (n=4370)** | | | | | |
| Number of days of care | 59 440 | 83 399 | 141 630 | 7345 | 4 |
| Number of babies receiving care | 4298 | 4050 | 4097 | 2268 | 1 |
| Mean (SD) duration of care (days)† | 13.6 (13.4) | 19.1 (19.5) | 32.4 (16.5) | 1.7 (3.0) | 0.0 (0.1) |
| 95% CI | 13.5 to 13.7 | 18.9 to 19.2 | 32.2 to 32.6 | 1.6 to 1.7 | 0.0 to 0.0 |
| Median (range) duration of care (days)† | 11.0 (0.0 to 246.0) | 14.0 (0.0 to 203.0) | 33.0 (0.0 to 127.0) | 1.0 (0.0 to 44.0) | 0.0 (0.0 to 4.0) |
| **29 weeks gestation (n=5036)** | | | | | |
| Number of days of care | 47 728 | 57 613 | 165 090 | 8449 | 0 |
| Number of babies receiving care | 4774 | 4631 | 4868 | 2671 | 0 |
| Mean (SD) duration of care (days)† | 9.5 (10.2) | 11.4 (14.1) | 32.8 (13.8) | 1.7 (3.0) | 0.0 (0.0) |
| 95% CI | 9.4 to 9.6 | 11.4 to 11.5 | 32.6 to 32.9 | 1.6 to 1.7 | 0.0 to 0.0 |
| Median (range) duration of care (days)† | 8.0 (0.0 to 258.0) | 7.0 (0.0 to 196.0) | 33.0 (0.0 to 119.0) | 1.0 (0.0 to 45.0) | 0.0 (0.0 to 0.0) |
| **30 weeks gestation (n=6625)** | | | | | |
| Number of days of care | 36 599 | 51 479 | 199 283 | 11 428 | 3 |
| Number of babies receiving care | 5196 | 5724 | 6471 | 3582 | 1 |
| Mean (SD) duration of care (days)† | 5.5 (7.1) | 7.8 (11.1) | 30.1 (12.1) | 1.7 (2.9) | 0.0 (0.0) |
| 95% CI | 5.5 to 5.6 | 7.7 to 7.8 | 29.9 to 30.2 | 1.7 to 1.8 | 0.0 to 0.0 |
| Median (range) duration of care (days)† | 4.0 (0.0 to 133.0) | 5.0 (0.0 to 213.0) | 30.0 (0.0 to 115.0) | 1.0 (0.0 to 37.0) | 0.0 (0.0 to 3.0) |
| **31 weeks gestation (n=8827)** | | | | | |
| Number of days of care | 29 094 | 44 547 | 219 355 | 14 948 | 9 |
| Number of babies receiving care | 5124 | 7055 | 8672 | 4718 | 2 |
| Mean (SD) duration of care (days)† | 3.3 (6.7) | 5.1 (7.5) | 24.9 (10.5) | 1.7 (2.8) | 0.0 (0.1) |
| 95% CI | 3.3 to 3.3 | 5.0 to 5.1 | 24.7 to 25.0 | 1.7 to 1.7 | 0.0 to 0.0 |
| Median (range) duration of care (days)† | 1.0 (0.0 to 201.0) | 3.0 (0.0 to 154.0) | 24.0 (0.0 to 240.0) | 1.0 (0.0 to 36.0) | 0.0 (0.0 to 7.0) |

CI: parametric confidence interval using Poisson distribution.
*19 babies had missing gestational age information.
†Estimated across all babies within a gestational age group.
HRG, healthcare resource group.

accounted for the largest proportion of days spent on the neonatal unit.

Online supplemental table A4 presents the number of babies who underwent hospital transfers and received key non-routine procedures, again by gestational age at birth. Greater resource utilisation among infants born at earlier gestations can be seen.

## Costs

The mean (SD) cost per baby for the various levels of care and non-routine procedures are shown by gestational age at birth in table 3 for the complete case analysis. As expected, and given the observations for resource use, costs can be seen to increase as gestational age at birth decreases. Mean (SD) total costs of routine care for a baby born at 27 weeks, for example, were, at £75 594 (£34,874), 2.8 times greater than costs for a

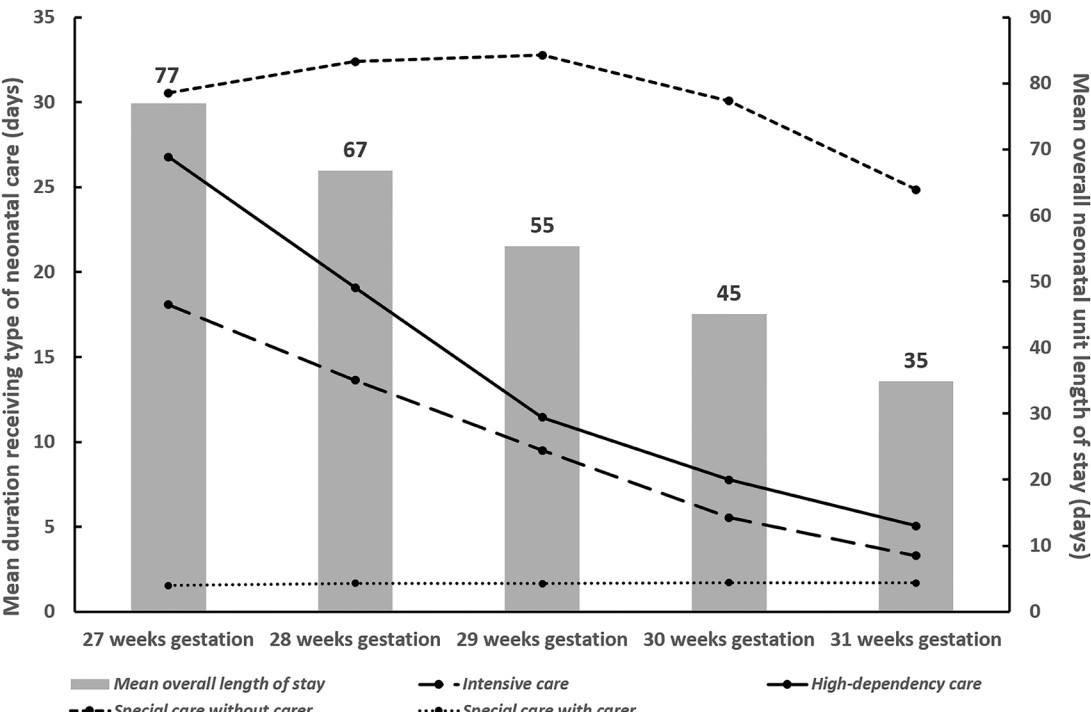

**Figure 1** Mean durations of different levels of daily care provided and overall length of stay per baby (in days) during neonatal unit admissions in England for the period 2014–2018. Data shown by gestational age at birth.

baby born at 31 weeks (£27 401 (£14,974)). The final column in table 3 shows that for the 2014–2018 cohort as a whole, just over 50% of total costs were attributable to the provision of intensive and high dependency care, with a further 42% coming from the provision of special care (without a carer). Further exploration of the composition of total costs in figure 2 reveals variation by gestational age at birth. Almost 70% of total costs for babies born at 27 weeks were associated with the use of intensive-dependency and high-dependency care, while the proportion decreased to only 36% for babies born at 31 weeks. The usage, and, thus, cost of special care (without a carer present) increases with increasing gestational age at birth. The results of the multiple imputation analysis for missing level of care costs (online supplemental table A5) were similar to the complete case analysis. Across all gestational age at birth groups, non-routine procedures accounted for only a very small proportion of total costs.

The overall total cost estimated for the complete case cohort (n=28 154) over the 5 years from 2014 to 2018 (£1.3 billion in table 3) suggests the annual costs of neonatal care for babies born between 27 and 31 gestational weeks and admitted to a neonatal unit in England to be around £262 million, with the provision of routine daily care on these units accounting for 96% (£252 million) of overall costs. Using the overall sample of 29 842 infants would increase the total cohort costs and the estimates of annual total costs shown in table 3 from £1 309 192 377 to £1 394 308 706 and the annual total cost from £262 million to £279 million.

## DISCUSSION
### Main findings

Analyses at the level of the individual baby revealed the main cost drivers to be the daily neonatal care provided to support babies. Analyses by gestational age at birth provided further evidence of the previously reported inverse relationship between resource use and healthcare costs and the degree of prematurity, with care for a baby born at 27 weeks estimated to cost almost three times more than for a baby born at 31 weeks.[5 7] Our work also provides valuable information on the contribution of different resource components to overall costs and illustrates how the mix of intensity of care required by these babies varies with gestational age at birth.

A number of studies have previously estimated neonatal care costs for preterm babies across a range of gestational ages.[2–7] One of the most recent by Rios et al in Canada[6] also used individual patient-level resource use data from a neonatal database (the Canadian Neonatal Network Database) and included around 8000 babies born between 27 and 31 weeks of gestation. A comparison between the costs estimated for the babies in the Rios et al study and the costs presented here (currency conversion from Can $ to UK £ made using Purchasing Power Parities) revealed the Canadian cost estimates to be consistently lower across all gestational age groups.[11] The most obvious explanation for these differences lies with the scope of the neonatal databases used by each study and the resulting implications for duration of neonatal unit stay. In general, Rios et al reported consistently lower mean lengths of stay for all gestational ages

**Table 3** Mean (SD) per baby and total neonatal care costs (2018/19 UK £) for very preterm babies in England over the period 2014–2018 using complete case analysis (n=28 154)*

| Resource use item | 27 weeks gestation (n=3296) Mean (SD) cost per baby | 28 weeks gestation (n=4370) Mean (SD) cost per baby | 29 weeks gestation (n=5036) Mean (SD) cost per baby | 30 weeks gestation (n=6625) Mean (SD) cost per baby | 31 weeks gestation (n=8827) Mean (SD) cost per baby | Total neonatal care costs (percentage) for 2014–2018 |
|---|---|---|---|---|---|---|
| Level of daily care | | | | | | |
| Intensive care | £27 690 (£24 039) | £20 865 (£20 473) | £14 548 (£15 603) | £8482 (£10 887) | £5062 (£10 235) | £356 747 496 (27.3%) |
| High-dependency care | £26 956 (£22 157) | £19 221 (£19 584) | £11 523 (£14 203) | £7828 (£11 186) | £5083 (£7,567) | £327 832 878 (25.0%) |
| Special care without carer | £20 187 (£11 880) | £21 407 (£10 921) | £21 659 (£9,110) | £19 875 (£8,019) | £16 423 (£6,939) | £546 285 433 (41.7%) |
| Special care with carer | £761 (£1,235) | £826 (£1,454) | £824 (£1453) | £847 (£1444) | £831 (£1378) | £23 230 160 (1.8%) |
| Normal care | £0 (£0) | £0.47 (£31) | £0 (£0) | £0.23 (£19) | £0.52 (£40) | £8224 (0.0%) |
| Total level of care costs | £75 594 (£34 874) | £62 319 (£30 841) | £48 554 (£23 426) | £37 033 (£18 276) | £27 401 (£14 947) | £1 254 104 191 (95.8%) |
| Hospital transfers | £1120 (£1,435) | £886 (£1,277) | £712 (£1,148) | £566 (£997) | £423 (£860) | £18 660 165 (1.4%) |
| Non-routine procedures | | | | | | |
| Nitric oxide | £105 (£557) | £81 (£436) | £49 (£354) | £36 (£312) | £19 (£148) | £1 356 929 (0.1%) |
| Surfactant replacement | £1006 (£2,052) | £933 (£1,980) | £784 (£1,970) | £497 (£995) | £351 (£1,340) | £17 742 128 (1.4%) |
| TPN>14 days use | £930 (£2,203) | £609 (£1,838) | £316 (£1,398) | £147 (£883) | £71 (£550) | £8 919 898 (0.7%) |
| Palivizumab | £172 (£1188) | £152 (£1518) | £74 (£1007) | £40 (£716) | £16 (£467) | £2 022 625 (0.2%) |
| ROP surgery | £61 (£398) | £21 (£223) | £12 (£165) | £8.9 (£138) | £2.9 (£71) | £439 674 (0.03%) |
| Neonatal surgery | £456 (£2053) | £358 (£1844) | £206 (£1325) | £135 (£996) | £107 (£934) | £5 946 767 (0.5%) |
| Total non-routine procedures costs | £2730 (£4,594) | £2153 (£4,228) | £1442 (£3,333) | £864 (£2,143) | £567 (£1955) | £36 428 021 (2.78%) |
| Total neonatal care costs for 2014–2018 | | | | | | £1 309 192,377 |
| Annual total neonatal costs | | | | | | £261 838 475 |

*19 babies had missing gestational age information.
NHS, National Health Service; ROP, retinopathy of prematurity; TPN, total parenteral nutrition.

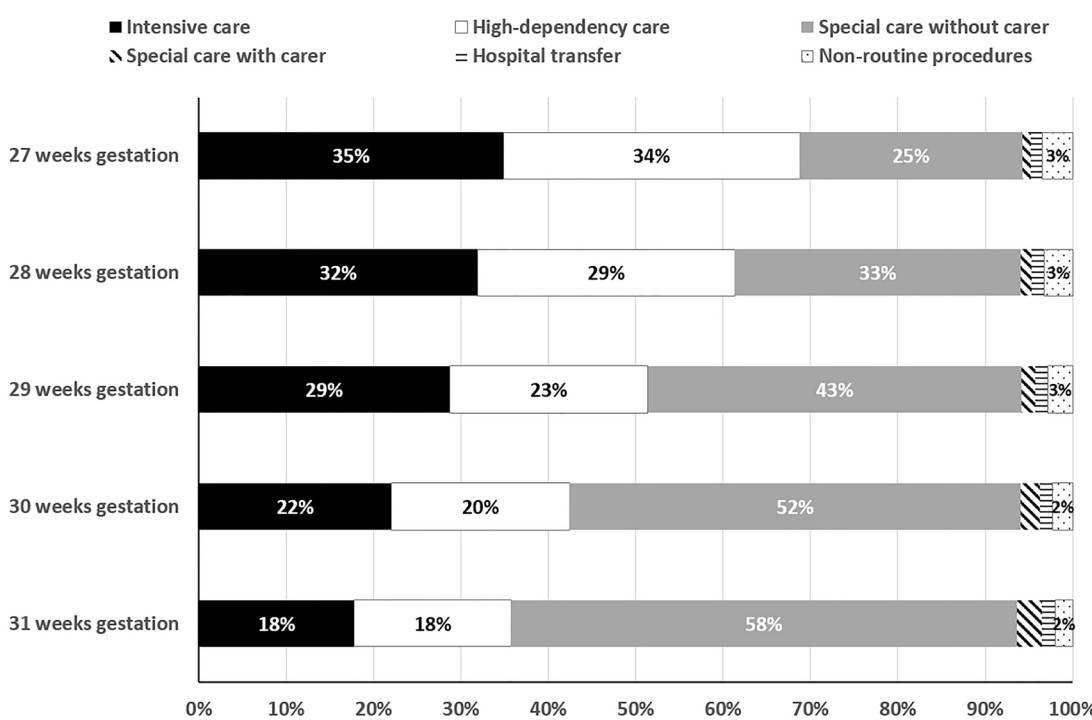

**Figure 2** Proportion of neonatal care cost attributable to different cost categories, by gestational age at birth.

compared with our estimates due to their focus on intensive care units and the exclusion of the costs of care in lower dependency neonatal units. Thus, the Rios *et al* estimates do not fully capture the true costs of healthcare provision for this group of babies.

In 2009, Mangham and colleagues used a decision analytic model to estimate neonatal costs for a hypothetical cohort of preterm babies born in England and Wales.[2] Model parameters were sourced from various cohort studies and the mean cost for a very preterm baby (<33 weeks) was estimated to be £57 726 (95% CI 28 779 to 94 868) (2006 GBP). With that analysis using a different methodology, reporting results for a wider gestational age range and now being well over a decade old, comparisons with the estimates reported here are challenging. While a number of more recent UK studies have been published on the costs of preterm birth, these works have focused specifically on births at other gestations (ie, moderate and late preterm births or extremely preterm births).[2 4 12 13] One reassuring finding, however, is that the costs of neonatal care for very preterm births presented here fall between the costs reported for moderate/late preterm infants and for extremely preterm infants in the UK.

**Strengths and limitations**

This study makes a number of contributions to the published literature. First, existing UK cost estimates for neonatal care following very preterm birth are over a decade old and so are unlikely to reflect current standards of practice.[2] This study provides up-to-date figures based on a large cohort of more than 28 000 babies who were discharged from or died within neonatal units across England between 2014 and 2018. Second, previous UK cost studies in the area have mainly relied on data synthesised from secondary sources to estimate costs. In utilising the NNRD, this study has been able to employ detailed and quality assured individual participant data derived from EPR. Furthermore, the data set offered a unique opportunity to capture near population-wide data (and thus costs) on day-to-day care provided across all neonatal units without imposing any additional burden on study participants. Third, the richness of the data set permitted us not only to cost routine daily care provided at differing levels of intensity but also to consider the cost implications of a number of major non-routine procedures, which are not captured within the HRG codes for neonatal critical care. These detailed data will help inform the planning and provision of care for very preterm babies. Finally, by generating up to date estimates of the mean resource use and costs of neonatal care for very preterm babies in England, this study provides valuable data of interest to a range of stakeholders, including NHS managers, clinicians providing care, researchers assessing the economic implications of therapies and interventions to prevent and treat preterm birth, and decision-makers charged with implementing new policies and allocating resources. The estimates are also likely to be informative for other countries with levels of neonatal care provision that are similar to England.

A number of limitations must also be acknowledged. This study considered only healthcare costs associated with the initial period of hospitalisation, though the economic consequences of preterm birth extend over prolonged periods of time, in some cases over the

lifetime of the individual.[2 13 14] Preterm birth can lead to additional healthcare as well as social care needs and special educational needs throughout childhood.[15 16] Further research is needed to adequately capture these wider costs accurately, including the economic costs to families while their preterm baby was hospitalised, and then throughout their lives. A further limitation is the exclusion from the analysis of babies with missing data on daily care provision, who were shown to have been born at earlier gestations than babies with complete data (online supplemental table A2). While these babies accounted for only 6% of the initial NNRD cohort and data showed no differences in gestational age at birth between the babies with complete data and all very preterm births registered in England (online supplemental table A4), we observed small significant differences in baseline characteristics between babies with missing and complete data. We conducted a multiple imputation approach to understand the implications of these differences and our results suggested limited impact of the missing data in the overall cost results.

## CONCLUSION

This study has generated up-to-date estimates of the costs of providing neonatal care to very preterm babies in England. Resource use and costs increase as gestational age decreases. The outputs from this work can be used to inform clinical and budgetary service planning and ensure the efficient allocation of healthcare resources. The estimates will also be of interest to countries with neonatal care provision similar to England.

**Author affiliations**
[1]National Perinatal Epidemiology Unit (NPEU), Nuffield Department of Population, University of Oxford, Oxford, UK
[2]Centre for Guidelines, National Institute for Health and Care Excellence, London, UK
[3]Faculty of Science and Engineering, University of Wolverhampton, Wolverhampton, UK
[4]Department of Neonatology, Leicester Royal Infirmary, University Hospitals Leicester NHS Trust, Leicester, UK
[5]Department of Population Health Sciences, University of Leicester, Leicester, UK
[6]Section of Neonatal Medicine, School of Public Health, Chelsea and Westminster Hospital Campus, Imperial College London, London, UK

**Acknowledgements** To parents, patients and families participating in the OPTI-PREM study, and to the OPTI-PREM Parent Panel, for its support. The OPTI-PREM study is grateful to its NIHR Project Steering Committee: Andrew Ewer (chair), Stavros Petrou, Gillian Santorelli, Josie Anderson, David Loughton, Lisa Stanton, Karen Luyt, and to the National Neonatal Collaborative. We are indebted to all members of the UK Neonatal Collaborative that participated and made this study possible (see supplementary file for list of members). The OPTI-PREM Study team: Elaine M Boyle, Neena Modi, Oliver Rivero-Arias, Bradley Manktelow, Sarah E Seaton, Natalie Armstrong, Miaoqing Yang, Abdul Qader T Ismail, Vasiliki Bountziouka, Caroline S Cupit, Alexis Paton, Victor L Banda, Elizabeth S Draper, Kelvin Dawson and Thillagavathie Pillay (Chief Investigator).

**Contributors** OR-A developed the study concept and aims, and developed the study proposal with MY, TP, EMB and NM. MY and ORA conducted the statistical analyses. MY, HC and OR-A wrote the first draft of the manuscript. All other authors reviewed the draft and provided critical input. All authors approve the final version. OR-A acts as the guarantor.

**Funding** This work is supported by the National Institute for Health Research, Health Services and Delivery Research Stream, project number 15/70/104 CRN accrual was approved by the NIHR for the period (1 August 2017 to 31 August 2018).

**Competing interests** No, there are no competing interests.

**Patient and public involvement** Patients and/or the public were involved in the design, or conduct, or reporting, or dissemination plans of this research. Refer to the Methods section for further details.

**Patient consent for publication** Not applicable.

**Ethics approval** Research ethics approval for the OPTI-PREM programme of work was obtained through the National Integrated Research Application System (IRAS, reference number 212 304 and research ethics committee reference number 17/NE/0800; North East—Tyne and Wear South).

**Provenance and peer review** Not commissioned; externally peer reviewed.

**Data availability statement** The authors do not have permission to supply the data used in this study. Data for this study was extracted from the National Neonatal Research Database (NNRD). The National Neonatal Research Database can be accessed by making a request to the Neonatal Data Analysis Unit at Imperial College London through the Health Data Research UK Gateway (https://web.www.healthdatagateway.org/search?search=NNRD&datasetSort=latest&tab=Datasets).

**ORCID iDs**
Helen Campbell http://orcid.org/0000-0003-2070-7794
Thillagavathie Pillay http://orcid.org/0000-0002-4159-3282
Oliver Rivero-Arias http://orcid.org/0000-0003-2233-6544

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
