## [Reviewer comments · BMJ Paediatrics Open]

ARTICLE DETAILS

TITLE (PROVISIONAL)	Neonatal health care costs of very preterm babies in England: a retrospective analysis of a national birth cohort
AUTHORS	Yang, Miaoqing Campbell, Helen Pillay, Thillagavathie Boyle, Elaine M. Modi, Neena Rivero-Arias, Oliver

VERSION 1 - REVIEW

REVIEWER	Reviewer Name: Dr. Asaph Rolnitsky Institution and Country: Sunnybrook Health Sciences Centre, Canada
REVIEW RETURNED	09-Jan-2023

GENERAL COMMENTS	Thank you for allowing me to contribute to this manuscript. This is a well structured descriptive analysis of neonatal costing of moderate-very preterm infants in UK. I do not review language and grammar. My only comments for this manuscript: 1. the need to simplify the tables and to explain the care types, as they are not typical of other jurisdictions2. as the distributions or costs are not normal, I suggest to emphasize the medians and ranges, preferably in a visual manner3. I suggest to explain the cost codes and their meaning, their calculations4. I would add more on other available cost studies for comparison.5. the text need a bit more to prove applicability to other sites. Overall, good work and important contribution to health services research. Good luck
---

REVIEWER	Reviewer Name: Dr. Emmanouil Bagkeris Institution and Country: University College London, Great Ormond Street Institute of Child Health, United Kingdom of Great Britain and Northern Ireland
REVIEW RETURNED	28-Jan-2023

GENERAL COMMENTS	A well written paper. Below are my recommendations: 1. In the methods section it is specified "Babies missing daily intervention records or for whom daily records were available but level of care data had not been recorded, were excluded from the analysis". Were those with missing data different in terms of gestational age, ethnicity, parity, maternal age, hospital from those
---

	with available data? Was imputation of data considered? The exclusion of these babies has been raised as a limitation of the study. Perhaps consider a sensitivity analysis with imputing the missing data and append as supplementary material the findings. This could aid the robustness of the results presented. 2. In the resource use section, you state “A clear trend for greater resource utilisation amongst infants born at earlier gestations can be seen.”. Consider quantifying and illustrating this trend using the appropriate correlation coefficient (Pearson's or Spearman's). 3. In table A4 percentages are reported in 1 decimal place while all other tables with percentages (Table 1, A2, A3) report 2 decimal places. If there are no precision concerns stick to one decimal place. 4. When STATA gives a p-value of 0.000 always report it as $p < 0.001$ 5. In table 2 the 95% CIs appear to be wrongly calculated. (e.g. the 95% CI for children 27 weeks gestation admitted for intensive care is $18.09 \pm 1.96 \times (15.70 / \sqrt{59494})$ gives a 95% CI of (17.96, 18.22) not (17.94, 18.23). Similar inconsistencies exist for children born at 29 weeks of gestation, 30 and 31. Please cross check the calculation of all 95% CIs reported. 6. In the abstract and manuscript, you chose to report the mean and SD of the cost. However, cost is not normally distributed. The 95% range contains non plausible values (e.g. table 3, for children attending intensive care born at 27 weeks of gestation the 95% range would be $27.690 \pm 1.96 * 24.039$ and this is (-19.426, 74.806)). When data are not normally distributed the median and IQR are the appropriate measures of centre and spread to be reported. I understand that previous publications/reports have used means and SDs to report the cost but this is not recommended.
--	---

VERSION 1 – AUTHOR RESPONSE

Neonatal health care costs of very preterm babies in England: a retrospective analysis of a national birth cohort

We would like to thank the editors as well as the two reviewers for their constructive and helpful feedback and comments. We have responded below to each of the comments and have indicated the relevant changes that are also highlighted in the revised manuscript using track changes.

Editor in Chief Comments:

Abstract Conclusions delete the 2nd sentence as it is not a conclusion and journal policy is for authors to avoid describing their study as the first or largest

Author's response: we have amended this sentence so there is no mentioning of first or largest in the abstract conclusion.

What this study adds delete the 1st sentence as it is Methods

Author's response: the sentence has been removed.

Introduction last sentence delete ", and so is, to the best of our knowledge, the largest participant-level study on the neonatal costs of very preterm babies in England" journal policy is for authors to avoid describing their study as the first or largest

Author's response: the sentence has now been deleted.

Discussion delete the first two sentences -journal policy is for authors to avoid describing their study as the first or largest Conclusion 1st sentence delete "the first to"

Author's response: these sentences have now been deleted.

Reviewer: 1

My only comments for this manuscript:

1. the need to simplify the tables and to explain the care types, as they are not typical of other jurisdictions

Author's response: We acknowledge that the levels of care will differ by geographical location and have now provided the reader with a reference to the NHS England web page which details each of the level of care types and the interventions they include. We have added the following sentence with reference to the final paragraph of page 5: "Detailed information on the items included within each daily level of care cost are available from NHS England."

We have also simplified Table 2 in the manuscript by reducing the number of separate reporting lines and reducing the number of decimal places reported. Re-spacing of the table now makes it easier to read. We were not able to further simplify Table 3 as this includes a required itemised break down of the cost estimates, however we have re-spaced the table making it easier to read.

2. as the distributions or costs are not normal, I suggest to emphasize the medians and ranges, preferably in a visual manner

Author's response: Healthcare cost data are usually not normal and this reflects the reality of the resources consumed by patients. In our study, the long right tails of cost distribution for the different categories of resource use captured babies that spend substantial amount of time in neonatal care because they required such level of care. They do not represent outliers that we know affects arithmetic means and distort clinical outcomes if not excluded. Focusing or emphasising cost medians will deviate the attention to the reader and provide and underestimation of the real costs. It is widely accepted in health economics that is the estimated population mean cost that is the statistic of interest to policy makers (Arrow 1970). Medians can provide useful descriptive information for resource use (Briggs 1998) (Mihaylova 2010) and we have provided mean ranges of duration of care in days in Table 2 for the benefit of the reader. However, we cautionary prefer to report arithmetic means for the cost results.

Arrow, K.J. and Lind, R.C. Uncertainty and the Evaluation of Public Investment Decisions. *The American Economic Review*, 1970. 60(3): 364-378.

Briggs, A. and Gray, A. The distribution of health care costs and their statistical analysis for economic evaluation. *Journal of Health Services Research & Policy*, 1998. 3(4): 233-245.

Mihaylova, B., Briggs, A., O'Hagan, A., and Thompson, S.G. Review of statistical methods for analysing healthcare resources and costs. *Health Econ*, 2011. 20(8): 897-916.

3. I suggest to explain the cost codes and their meaning, their calculations

Author's response: We have added the following text to the final paragraph on page 5, which describes how the HRG codes are devised, what they are used for, and how they are costed to generate the bed day cost estimates used in the analysis.

"Such costs are estimated by assigning each level of care a healthcare resource group (HRG) code. HRGs are groupings of clinically meaningful activities made primarily on the basis of diagnosis and procedure codes, and within the NHS, are the 'units' of health care for which providers receive payment. Costs assigned to each HRG are based upon nationally estimated tariffs developed to adequately cover the cost of providing high quality and cost-effective care."

4. I would add more on other available cost studies for comparison.

Author's response: We have added additional text to the discussion section (final paragraph, page 9) further discussing the UK-based study conducted on the costs of pre-term birth and why making comparisons between that study and our own study is challenging.

"In 2009, Mangham and colleagues used a decision analytic model to estimate neonatal costs for a hypothetical cohort of preterm babies born in England and Wales. Model parameters were sourced from various cohort studies and the mean cost for a very pre-term baby (<33 weeks) was estimated to be £57,726 (95% CI: 28,779 to 94,868) (2006 GBP). With that analysis using a different methodology, reporting results for a wider gestational age range, and now being well over a decade old, comparisons with the estimates reported here are challenging."

5. the text need a bit more to prove applicability to other sites.

Author's response: we refer the reviewer to our response on the first point.

Reviewer 2

1. In the methods section it is specified "Babies missing daily intervention records or for whom daily records were available but level of care data had not been recorded, were excluded from the analysis". Were those with missing data different in terms of gestational age, ethnicity, parity, maternal age, hospital from those with available data? Was imputation of data considered? The exclusion of these babies has been raised as a limitation of the study. Perhaps consider a sensitivity analysis with imputing the missing data and append as supplementary material the findings. This could aid the robustness of the results presented.

Author's response: We reported in the supplementary Table A2, the baseline characteristics of babies born with and without missing daily records or daily care provided. We observed significant differences in some of the characteristics but some of the differences were likely driven by the large sample size we are employed and not necessarily by real clinical differences. Therefore, our hypothesis was that imputation was not needed. However, we have revisited this hypothesis and added a secondary analysis of the level of care cost data (main driver of cost results) using multiple imputation as suggested by the reviewer. The results of this new analysis is presented in Table A5. We have made several changes in the methods, results and discussion to reflect this new information. We thank the reviewer for this comment as we think it has strengthen our story.

2. In the resource use section, you state "A clear trend for greater resource utilisation amongst infants born at earlier gestations can be seen.". Consider quantifying and illustrating this trend using the appropriate correlation coefficient (Pearson's or Spearman's).

Author's response: We have removed the statement "clear trend" from the sentence as we do not think such information is needed to convey the message that babies born earlier consumed more resources.

3. In table A4 percentages are reported in 1 decimal place while all other tables with percentages (Table 1, A2, A3) report 2 decimal places. If there are no precision concerns stick to one decimal place.

Author's response: we have revised all percentages and report only 1 decimal place except in Table 3 where two decimals were needed to distinguish proportions clearly and to present the normal care costs figures.

4. When STATA gives a p-value of 0.000 always report it as $p < 0.001$

Author's response: we have amended this in Table A2.

5. In table 2 the 95% CIs appear to be wrongly calculated. (e.g. the 95% CI for children 27 weeks gestation admitted for intensive care is $18.09 \pm 1.96 \times (15.70 / \sqrt{59494})$ gives a 95% CI of (17.96, 18.22) not (17.94, 18.23). Similar inconsistencies exist for children born at 29 weeks of gestation, 30 and 31. Please cross check the calculation of all 95% CIs reported.

Author's response: we think the reviewer has used the incorrect sample to estimate the confidence interval as it is the number of babies and not the total number of days of care received that should be used. However, we missed to report that the confidence intervals in Table 2 were calculated parametrically using a Poisson distribution and not a normal distribution. We have clarified this now in the statistical analysis in the methods section and Table 2.

6. In the abstract and manuscript, you chose to report the mean and SD of the cost. However, cost is not normally distributed. The 95% range contains non plausible values (e.g. table 3, for children attending intensive care born at 27 weeks of gestation the 95% range would be $27.690 \pm 1.96 * 24.039$ and this is (-19.426, 74.806)). When data are not normally distributed the median and IQR are the appropriate measures of centre and spread to be reported. I understand that previous publications/reports have used means and SDs to report the cost but this is not recommended.

Author's response: we would like to note that Table 3 reports standard deviations (SD) and not standard errors (SE) and therefore the calculation by the reviewer is incorrect. The correct calculation is $27690 \pm 1.96 * (24039 / \sqrt{3296})$ and the confidence interval is (26869, 28511). We understand the point made that for cost data close to zero normal parametric approaches to confidence intervals may not be appropriate. However this is not the case in Table 3. We refer the reviewer to our response to the second point raised by the first reviewer about reporting means instead of medians for cost data.

VERSION 2 – REVIEW

REVIEWER	Reviewer Name: Dr. Asaph Rohnitsky Institution and Country: Sunnybrook Health Sciences Centre, Canada
REVIEW RETURNED	28-Mar-2023

GENERAL COMMENTS	Thank you for your submission and for allowing me to contribute to the success of this submission. This is an analysis of cost of care for moderately preterm infants in UK, at the payor level. The methods and analysis, discussion, are sound and clear. My only comments (I do not review grammar etc): 1. Along the manuscript it sometimes seems like the the costs are DAILY (per day) and thus incredibly high. I suggest to rephrase it to be clear that it's the total stay as calculated by daily costs.
---

	2. I'd add more on the calculation methods and the addition of the subcosts in the methods, as they are not detailed clearly. 3. More recent studies cost the NICU care of infants in Canada (Rolnitsky, Bell et al.) and I'd advise to add them to the discussion. 4. I'd add more to the discussion on the implication of the study, applicability to other jurisdictions, and potential benefits to patients and the public. I hope you find those comments useful. Thank you and good luck
--	--

VERSION 2 – AUTHOR RESPONSE

Neonatal health care costs of very preterm babies in England: a retrospective analysis of a national birth cohort

We would like to thank the two reviewer for their constructive and helpful feedback and comments. We have responded below to each of the comments and have indicated the relevant changes that are also highlighted in the revised manuscript using track changes.

Reviewer's comments

1. Along the manuscript it sometimes seems like the costs are DAILY (per day) and thus incredibly high. I suggest to rephrase it to be clear that it's the total stay as calculated by daily costs.

Author response: We are grateful to the reviewer for drawing our attention to this and have amended the wording in the abstract and the results section (paragraph 2, page 8) to make it clearer that we report the costs of neonatal unit stay at each level of care, as opposed to the daily cost of care.

2. I'd add more on the calculation methods and the addition of the subcosts in the methods, as they are not detailed clearly.

Author response: We have added additional text to the methods section (paragraph 1, page 6) on how we costed the high cost non-routine procedures.

3. More recent studies cost the NICU care of infants in Canada (Rolnitsky, Bell et al.) and I'd advise to add them to the discussion.

Author response: We now cite the additional study mentioned by the reviewer as one of several estimating the costs of pre-term births, in both the introduction and discussion sections of the paper.

4. I'd add more to the discussion on the implication of the study, applicability to other jurisdictions, and potential benefits to patients and the public.

Author response: We have added sentences to the discussion section (paragraph 2, page 10) which note that the costs we have presented will be informative for other countries with neonatal care provision that is similar to England, and that the results we have reported will help with the planning and provision of care for very pre-term babies.